# Clinical outcomes and associated factors among mechanically ventilated patients in adult intensive care unit at Jimma Medical Center, Southwest Ethiopia, 2024

Abera Mulatu Uma[1]*, Shemsadin Ame[2], Segni Gemechu Hurisa[3], Mulu Kitaba Negawo[4], Sheka Shemsi Seid[2]

1 Department of Emergency and Critical care Nursing, School of Nursing and Midwifery, College of Health Science and Referral hospital, Ambo University, Ambo, Ethiopia, 2 Department of Emergency and Critical care Nursing, School of Nursing, Institute of Health, Jimma University, Jimma, Ethiopia, 3 Department of Medical Physiology, School of Medicine, College of Health Science and Referral hospital, Ambo University, Ambo, Ethiopia, 4 Department Nursing, School of Nursing and Midwifery, College of Health Science and Referral hospital, Ambo University, Ambo, Ethiopia

* aberamulatu1212@gmail.com

## Abstract

### Background

Mechanical ventilation (MV) is a life-saving intervention for patients unable to maintain adequate oxygenation or ventilation. However, MV is also associated with complications such as ventilator-associated pneumonia, lung injury, and prolonged ICU stay, which may increase morbidity and mortality. Previous studies in Ethiopia and the region are limited and do not fully address key predictors of clinical outcomes among mechanically ventilated patients. Objective: To assess clinical outcomes and associated factors among mechanically ventilated patients in adult intensive care units at Jimma Medical Center, Southwest Ethiopia, and in2024.

### Methods

A retrospective cross-sectional study was conducted among 411 randomly selected medical records of mechanically ventilated adult patients. Data were extracted using a structured and pretested tool. Statistical analysis included bivariable and multivariable logistic regression. Variables with $p < 0.25$ were entered into the multivariable model, and statistical significance was declared at $p < 0.05$. Adjusted Odds Ratios (AORs) with 95% Confidence Intervals (CIs) were reported.

### Results

Of the 411 reviewed patient charts, 215 (52.3%) were male with a mean age of $36.9 \pm 16.2$ years. The overall mortality rate was 54% (95% CI: 49.4–58.6). Significant predictors of mortality included hospital stay > 5 days before intubation

**Data availability statement:** All relevant data are within the paper and its Supporting Information files.

**Funding:** The author(s) received no specific funding for this work.

**Competing interests:** The authors declare that they have no competing interests.

**Abbreviations:** A/C, Assisted Control; AICU, Adult Intensive Care Unit; ARDS, Acute Respiratory Distress Syndrome; ARF-CPD, Acute Respiratory Failure-Chronic Pulmonary Disease; CHF, Cardiac Heart Disease; CKD, Chronic Kidney Disease; COPD, Chronic Obstructive Pulmonary Disease; CPAP, Continues Positive Airway Pressure; CPR, Cardiopulmonary Resuscitation; HIV/AIDS, Human Immune Variance/Acquire Immune Disease Syndrome; IMV, Invasive Mechanical Ventilation; LMIC, Low Middle-Income Country; MODS, Multiple Organ Deaths Syndrome; NMIV, Non-Invasive Mechanical Ventilation; PCV, Pressure Control Ventilation; PEEP, Positive End Expiratory Pressure; PSV, Positive Pressure Ventilation; SIMV, Synchronic Intermittent Mandatory Ventilation; TBI, Traumatic Brain Injury; TT, Triangle Tube; VAP, Ventilation Associated Pneumonia; VCV, Volume Control Ventilation.

(AOR = 3.03, 95% CI: 1.05–6.09), re-intubation (AOR = 9.02, 95% CI: 4.07–20.01), presence of comorbidities (AOR = 9.12, 95% CI: 4.38–18.99), development of complications (AOR = 6.84, 95% CI: 3.15–14.81), multi-organ dysfunction syndrome (MODS) (AOR = 3.48, 95% CI: 1.47–7.82), and use of sedation (AOR = 0.36, 95% CI: 0.18–0.74).

## Conclusion

Mechanically ventilated patients at Jimma Medical Center experience a high mortality rate. Key determinants of mortality were re-intubation, comorbidities, complications, MODS, and delayed intubation. Improving ICU protocols, early identification of high-risk patients, proper monitoring, and prevention of unplanned extubation may improve survival outcomes.

## Introduction

Mechanical ventilation (MV) is an essential life-support intervention used when patients cannot maintain adequate gas exchange independently [1]. Despite its life-saving benefits, MV is also associated with substantial risks, including ventilator-associated pneumonia, barotrauma, hemodynamic instability, and prolonged ICU stay that may worsen clinical outcomes in critically ill patients [2–4]. In low- and middle-income countries (LMICs), these risks are often exacerbated by shortages of trained ICU staff, limited equipment, and delays in critical care interventions [5,6].

Globally, the burden of patients requiring mechanical ventilation is increasing, largely due to rising trauma, infectious diseases, and non-communicable diseases (NCDs) [7]. However, in Ethiopia, research on MV outcomes remains limited, with most available studies conducted in few urban tertiary centers and lacking comprehensive assessment of key determinants such as re-intubation status, mode of ventilation, pre-intubation clinical status and accessibility of assurances [8–14].

Previous studies in Ethiopia have reported high mortality rates among mechanically ventilated patients, ranging from 41% to 57%, reflecting significant challenges in critical care delivery [2,8–10,14,15]. Yet these studies often lacked standardized definitions, did not sufficiently examine important variables like multi-organ dysfunction syndrome (MODS), and did not investigate prolonged pre-intubation hospitalization, a potentially modifiable factor.

Furthermore, the Adult Intensive Care Unit (AICU) at Jimma Medical Center (JMC) serves a large catchment population and frequently manages complex critically ill patients, making it an important setting for understanding MV outcomes. To date, no comprehensive study has examined clinical outcomes and associated factors among mechanically ventilated adult patients at JMC over a long period, thus creating a critical evidence gap for improving care service and policy.

Therefore, this study aimed to assess clinical outcomes and identify key determinants of mortality among mechanically ventilated adult patients at Jimma Medical Center from 2019 to 2023. By addressing previously unexamined variables

and synthesizing a 5-years ICU data, this study provides evidence that may improve ICU protocols, guide clinical decision-making, and inform resource allocation.

## Methods and materials

### Study Area

This study was conducted at Jimma Medical Center (JMC), located 352 km southwest of Addis Ababa, Ethiopia. JMC is the largest referral and teaching hospital in southwest Ethiopia and serves an estimated more than 20 million people. The hospital provides advanced critical care services across multiple intensive care units (ICUs), including emergency, surgical, and medical ICUs, all of which admit mechanically ventilated patients.

### Study period

The study utilized medical records of patients treated between January 1, 2019, and December 30, 2023, while data extraction was carried out from April 30 to May 30, 2024.

### Study design

An institution-based retrospective cross-sectional study design was employed. This design was intentionally chosen because it allowed the researcher for systematic extraction of detailed clinical information from existing medical records, assessment of outcomes and factors associate among mechanically ventilated patients over a five-year period.

### Population

**Source population.** All mechanically ventilated adult patients admitted to the Adult Intensive Care Unit (AICU) at JMC during the study period.

### Study population

Randomly selected medical records of adult patients who received mechanical ventilation at JMC between 2019 and 2023 constituted the study population.

### Inclusion and exclusion criteria

**Inclusion criteria.** Illegible patient charts were considered for adult patients aged 18 years and above and received mechanical ventilation for ≥ 24 hours in the AICU during the study period [16].

**Exclusion criteria.** The exclusion criteria were patients referred to other hospitals before outcome determination, patients who left against medical advice and charts with missing essential variables required for classification of outcomes. Illegible or incomplete charts were excluded only when key variables, such as outcomes, major clinical characteristics, or ventilation details, could not be reliably extracted.

### Sample size determination

The sample size for the primary objective was determined using a single population proportion formula, assuming a 41.7% mortality rate from a previous Ethiopian study, with a 95% confidence level and 5% margin of error. The calculated sample size was **374**.

After adding a **10%** allowance for missing or incomplete charts, the final required sample size increased to **411**.

Additionally, key predictors reported in Ethiopian and international studies (e.g., re-intubation, sedation, mode of ventilation) were evaluated using Epi-Info STATCALC. All secondary calculations yielded smaller sample sizes than the primary estimate, supporting 411 as the final sample size.

## Sampling procedure

A sampling frame of all mechanically ventilated patients from emergency, surgical, and medical ICUs was prepared. Proportional allocation was performed based on the number of MV cases in each ICU and to ensure representativeness and minimized selection bias, simple random sampling using a computer-generated list was then used to select the final 411 records.

## Variables

### Dependent variable

**Clinical outcome (Survived / Died).**

### Independent variables

Socio-demographics (age, sex, residence)

Admission characteristics (diagnosis, GCS, vital signs)

Comorbidities

Mechanical ventilation parameters (type, mode, settings)

Pre-intubation hospital stay

Re-intubation

Complications developed

Sedation use

ICU length of stay

Length of stay in the hospital before intubation

**All abbreviations were expanded at first mention:**

Volume-controlled ventilation (VCV)

Pressure-controlled ventilation (PCV)

Synchronized intermittent mandatory ventilation (SIMV)

Continuous positive airway pressure (CPAP)

Multiple organ dysfunction syndrome (MODS)

### Operational definitions

**Prolonged pre-intubation hospitalization:** > 5 days in hospital before intubation [17,18].
   **Re-intubation:** Any second endotracheal intubation after prior extubation within the same admission [19].
   **Complication:** Any new clinical problem after initiation of MV (e.g., VAP, ETT obstruction, unplanned extubation) [20].
   **MODS:** Failure of ≥2 organ systems documented in the patient record.
   **Comorbidity:** Pre-existing chronic condition diagnosed before or during admission [21].
   **Sedation** was defined as having received an intravenous or intramuscular sedative (ketamine, propofol, ketofol, diazepam, and morphine) for any period during the intensive care stay [22]. This does not include the sedation for the procedure [23,24].

**Against medical advice** is a term used when a patient decides to leave the hospital or refuse a recommended treatment due to cost or the patient's preference [14].

**An incomplete chart** refers to a medical record that is unclear, vague, or lacking in completeness and that does not provide the patient's entire story is considered incomplete [25].

**Clinical outcomes** were determined based on the information documented in the records regarding the patient's condition at the time of discharge from the intensive care unit. Also for multivariable analysis, the patient outcome was categorized into two categories as appropriate (died or recovered (discharged alive)) [4,9].

### Data collection procedure

Data were collected using a structured and pretested extraction tool adapted from reviewed literature. The tool captured patient demographics, admission information, ventilation parameters, clinical conditions, complications and outcomes. Four trained ICU nurses performed data extraction after receiving training to ensure consistency.

### Data processing and analysis

Data were entered into **EpiData 4.6** and analyzed using **SPSS version 26**.

Bivariable logistic regression was performed first and variables with **$p < 0.25$** entered multivariable logistic regression. Statistical significance was declared at **$p < 0.05$**.

### Model diagnostics included

Hosmer–Lemeshow goodness-of-fit test and Variance Inflation Factor (VIF) for multicollinearity. Variables with $VIF > 10$ (e.g., temperature, pulse rate, liver enzymes) were excluded. The mean VIF for the remained variables was 1.38. Results were presented using tables, figures, and narrative summaries.

### Ethical considerations

Ethical approval was obtained from the Institutional Review Board (IRB) of Jimma University. Additionally, permission to access medical records was granted by the hospital administration. Because the study involved retrospective review of anonymized medical records, the requirement for individual informed consent was formally waived by the IRB. All extracted data were kept confidential, and no personal identifiers were recorded.

### Results

Soco-dimographic and Admission Characteristics among mechanically ventilated patients in the AICU at JMC.

A total of 1,233 patients were admitted and received mechanical ventilation (MV) in the AICU at JMC from January 1, 2019, to December 30, 2023. Out of these, 411 medical records were reviewed. Most patients (67.9%) were aged between 18 and 40 years, with a mean age of 36.9 ± 16.2 years, ranging from 18 to 94 years. More than half of the patients were male, 215 (52.3%) (See Fig 1).

The most common primary admission diagnoses were respiratory disorders (22%), followed by other infectious (18%) and Neurological Disorders (14%). At ICU admission, 52.3% had a Glasgow Coma Scale (GCS) score below 8, the most predominant initial indication of mechanical ventilation was respiratory failure 195 (47%), followed by GCS < 8 or coma 88 (21%). additionally; comorbidities were documented in 56.4% of patients, with hypertension and chronic heart disease being the most common;(see Table 1).

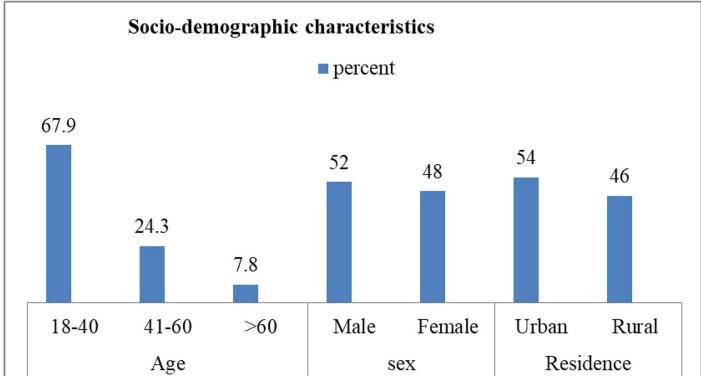

**Fig 1. Socio-demographic characteristics among mechanically ventilated patients in the AICU at JMC from 2019 to 2023.**

**Table 1. Indication of mechanical ventilators and related characteristics among mechanically ventilated in the AICU at JMC from 2019 to 2023 (n = 411).**

| Variables | Category | Frequency | Percent (%) |
|---|---|---|---|
| Variables | Category | Frequency | Percent (%) |
| Admission Diagnosis | Cardiovascular Disorders | 39 | 9.5 |
| | Gastrointestinal Disorders | 51 | 12 |
| | Neurological Disorders | 57 | 4.1 |
| | Obstetric and Gynecological Disorders | 34 | 8.3 |
| | Other infectious | 73 | 18 |
| | Trauma and Injuries | 52 | 13 |
| | Respiratory Disorders | 89 | 22 |
| | Miscellaneous | 16 | 3.9 |
| | Other | 37 | 9 |
| Indication for MV initiation | Respiratory failure | 195 | 47 |
| | Cardiac failure | 48 | 12 |
| | CNS failure | 17 | 4.1 |
| | Sepsis | 45 | 11 |
| | GCS < 8 | 88 | 21 |
| | Other | 18 | 4.4 |
| Existed Co-morbidity | Yes | 232 | 56.4 |
| | No | 179 | 43.6 |
| Types of comorbidity | HTN | 50 | 20 |
| | DM | 26 | 10 |
| | CHF | 74 | 28.8 |
| | CKD | 31 | 12.4 |
| | HIV | 16 | 6.3 |
| | CNS disorder | 12 | 4.7 |
| | neoplastic | 20 | 7.9 |
| | Others | 25 | 9.8 |

## Mechanical ventilation–related Factors among mechanically ventilated patients in the AICU at JMC

Concerning complications, more than half of 252 (61.3%) of patients had developed complications. Of these complications, VAP and bed sores accounted for 55 (21%) and 52 (20%), respectively. More than one-third, 138 (33.6%) of the patients were re-intubated, and ETT block was the predominant reason for the re-intubation, which accounted for 52 (12.5%). Concerning multi-organ dysfunction syndrome (MODS), 316 (76.9%) of the patients had MODS. The most prominent MODS were cardiac (36.7%), renal (25.3%), and hepatic failures (20.8%); (see Table 2).

**Table 2. Complications, re-intubation, and MODS-related factors among mechanically ventilated patients in the AICU at JMC from 2019 to 2023 (n = 411).**

| Variables | Category | Frequency | Percent (%) |
|---|---|---|---|
| Mode of initial ventilation | A/C(VCV) | 223 | 54.3 |
| | A/C(PCV) | 39 | 9.5 |
| | SIMV | 128 | 31.1 |
| | CPAP | 21 | 5.1 |
| Mode of airway access | ETT | 376 | 91.5 |
| | Tracheostomy | 21 | 5.1 |
| | CPAP mask | 14 | 3.4 |
| Developed complication | Yes | 252 | 61.3 |
| | No | 159 | 38.7 |
| Type of Complication | VAP | 55 | 21 |
| | Bed sore | 52 | 20 |
| | HAP | 47 | 15 |
| | Electricity's imbalances | 18 | 6.5 |
| | Pneumothoraxes | 30 | 11 |
| | Prolonged duration of ventilation | 59 | 21 |
| | post-extubated stridor's | 3 | 1.1 |
| | lung Atelectasis | 4 | 1.4 |
| | Others | 14 | 5.0 |
| Weaning tried? | Yes | 212 | 52 |
| | No | 199 | 48 |
| Weaning methods | SIMV | 32 | 15.2 |
| | CPAP | 134 | 63 |
| | PS with CPAP | 39 | 18 |
| | T-piece with PS | 8 | 3.8 |
| Method of extubation | Extubated after weaning | 331 | 80.5 |
| | self-extubated | 44 | 10.7 |
| | accidental extubated | 36 | 8.76 |
| Re-intubated? | Yes | 138 | 33.6 |
| | No | 273 | 66.4 |
| Cause of re-intubation | ETT block | 52 | 12.5 |
| | Accidental extubating during procedure | 31 | 7.5 |
| | Self-extubating | 33 | 8 |
| | Post extubating failure | 19 | 4.6 |

**Clinical and Hospital-Related Factors among mechanically ventilated patients in the AICU at JMC**

Out of ventilated patient 30.2% of them were developed multiple organ dysfunction syndrome (MODS). Among ventilated patients, cardiopulmonary resuscitation (CPR) was performed for (22.4%) of patients within 24 hours of ventilation. Of the ventilated patients, (62.5%) used Deep Vein Thrombosis (DVT) prophylaxis. The most prominent drug used for DVT prophylaxis was heparin 256 (62.3%). Of the total number of ventilated patients, 238 (58%) used inotropic drugs. The most commonly used inotropic drugs were adrenaline (48.9%), followed by dopamine (23.6%). Regarding sedation status of these patients 41.1% of cases were used sedation with particularly midazolam or fentanyl. The mean and standard deviation of the duration of illness before hospital admission were $17.2 \pm 15.9$ days, and the duration of hospital stay before intubation was $5.3 \pm 4.8$ days. Also, the mean length of stay on mechanical ventilator (MV), in the intensive care unit (ICU), and hospital were $10.1 \pm 11$, and $11 \pm 12.19$ days, respectively; (see Table 3).

**Table 3. Clinical and Hospital-Related Factors among mechanically ventilated patients in the AICU at JMC from 2019–2023 (n = 411).**

| Variables | Category | Frequency | Percent (%) |
|---|---|---|---|
| MODS developed | Yes | 316 | 76.9 |
| | No | 107 | 26 |
| Type of organ failure | Renal | 116 | 36.7 |
| | Cardiac | 80 | 25.3 |
| | Hepatic | 66 | 20.8 |
| | CNS failure | 35 | 11 |
| | Others | 19 | 6 |
| CPR within 24 hours of ICU admission | Yes | 92 | 22.4 |
| | No | 319 | 77.6 |
| DVT prophylaxis used? | Yes | 258 | 63 |
| | No | 153 | 37 |
| Type of DVT prophylaxis used | Heparin | 256 | 99.2 |
| | Warfarin | 2 | 0.8 |
| Inotropic used? | Yes | 238 | 57.9 |
| | No | 173 | 42.1 |
| Type of inotropic drug used | Adrenaline | 201 | 85 |
| | Noradrenaline | 11 | 5 |
| | Dopamine | 23 | 9.2 |
| | Others | 2 | 0.8 |
| Sedation used | Yes | 230 | 56 |
| | No | 181 | 44 |
| Type of sedation used | Ketamine | 53 | 22 |
| | Propofol | 32 | 13 |
| | Ketofol | 36 | 15 |
| | Diazepam | 67 | 28 |
| | Morphine | 50 | 21 |
| Insurance access | yes | 187 | 45.5 |
| | no | 224 | 54.5 |
| variable | **Mean** | **SD** | |
| Duration of illness before Hospital admission | 17 | 15.9 | |
| Duration of hospital stay before intubation | 5.3 | 4.8 | |
| Lengths of stay under MV in a day | 10.1 | 12 | |
| Lengths of stay in ICU in a day | 11 | 12.19 | |

### Clinical outcome among mechanically ventilated patients in the adult intensive care unit at Jimma Medical Center from 2019 to 2023

Out of the total number of mechanically ventilated patients, two hundred twenty-two (54%) with 95% CI (49.24, 58.6) died, whereas one hundred eight nine (46%) recovered and were discharged alive; (see Fig 2).

### Factors associated with mortality

**Bivairable analysis.** In bivairable analysis, many variables were significantly associated with mortality at p < 0.25 and therefore included in the multivariable model. These variables included, patient age, Pre-intubation hospital stay, re-intubation, presence of comorbidities, development of complications, sedation use, MODS, GCS of patients during admission, serum electrolytes, Creatine level, and CPR done;(see Table 4)

### Multivariable logistic regression

In the multivariable logistic regression model, the following variables remained statistically significant predictors of mortality at declared statistical significance

### Pre-intubation hospital stay > 5 days

Patients who stayed more than five days in the hospital before intubation were three times more likely to die compared to those with shorter stays (AOR = 3.03; 95% CI: 1.05–6.09).

### Re-intubation

This finding revealed re-intubated patients had significantly higher odds of mortality with **(AOR = 9.02; 95% CI: 4.07–20.01).**

### Presence of comorbidities

The analysis of this study also yielded presence of other comorbidities significantly increased the risk of death with **(AOR = 9.12; 95% CI: 4.38–18.99).**

### Development of complications

We also found patients who developed complications after initiation of MV had higher mortality compared to those recovered without developing complication with **(AOR = 6.84; 95% CI: 3.15–14.81).**

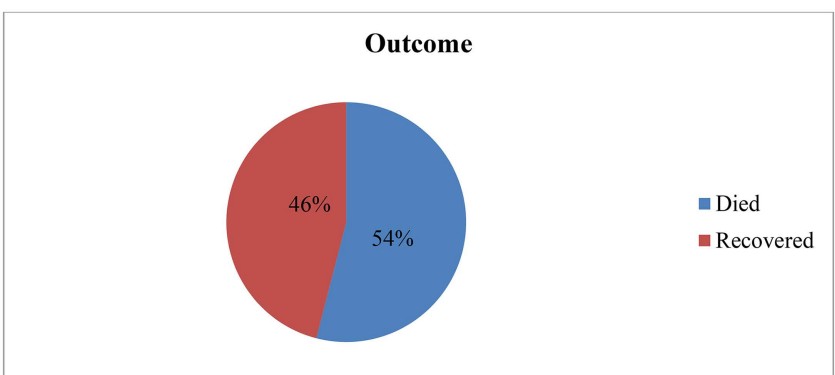

**Fig 2. Death outcome among mechanically ventilated patients in the AICU at JMC from 2019 to 20239 (n = 411).**

**Table 4. Result of binary and multivariable logistic regression analysis for associated factors with clinical outcome among mechanically ventilated patients in the adult intensive care unit at Jimma Medical Center from 2019 to 2023 (n = 411).**

| Variables | Category | | COR (95%CI) | AOR (95%CI) | P-values |
|---|---|---|---|---|---|
| | **Died** | **Recovered** | | | |
| GCS | | | | | |
| 3-8 | 138 | 77 | 3.465 (1.750-6.859) | 1.190 (0.352-4.020) | 0.780 |
| 9-12 | 69 | 83 | 1.607 (0.798-3.238) | 0.516 (0.146-1.817) | 0.303 |
| 13-15 | 15 | 29 | 1.00 | 1.00 | |
| LOS Hospital before intubate | | | | | |
| ≤5 days before intubate | 99 | 133 | 1.00 | 1.00 | |
| >5 days before intubate | 123 | 56 | 2.9511 (1.960-4.443) | 3.032 (1.509-6.091) | 0.002** |
| Re-intubation | | | | | |
| Yes | 103 | 35 | 3.808 (2.424-5.985) | 9.029 (4.073-20.018) | 0.001** |
| No | 119 | 154 | 1.00 | 1.00 | |
| Comorbidities | | | | | |
| Yes | 169 | 63 | 6.377 (4.140-9.824) | 9.121 (4.380-18.993) | 0.001** |
| No | 53 | 126 | 1.00 | 1.00 | |
| Complication | | | | | |
| Yes | 181 | 71 | 7.337 (4.683-11.495) | 6.840 (3.158-14.82) | 0.001** |
| No | 41 | 118 | 1.00 | 1.00 | |
| MODS | | | | | |
| Yes | 202 | 102 | 8.615 (5.014-14.801) | 3.477 (1.471-8.215) | 0.005** |
| No | 20 | 87 | 1.00 | 1.00 | |
| Hemoglobin level | | | | | |
| Hb<7 severe anemia | 57 | 28 | 2.288 (1.356-3.860) | 1.476 (0.577-3.774) | 0.416 |
| Hb 7–10.9 mg/dl moderate anemia | 60 | 43 | 1.568 (0.978-2.513) | 0.741 (0.318-1.731) | 0.489 |
| Hb 11.1–16.9 mg/dl | 105 | 118 | 1.00 | 1.00 | |
| CPR done within 24 hours of ICU admission | | | | | |
| Yes | 74 | 18 | 4.750 (2.713-8.317) | 7.942 (0.894-21.798) | 0.1 |
| No | 148 | 171 | 1.00 | 1.00 | |
| Serum k level | | | | | |
| >5.5mEq | 139 | 43 | 12.08 (6.557-22.253) | 6.527 (0.506-16.998) | 0.1 |
| <3.5mEq | 64 | 75 | 3.189 (1.739-5.847) | 2.042 (0.815-5.119) | 0.292 |
| 3.5-5.5mEq | 19 | 71 | 1.00 | 1.00 | |
| OFT of Creatine level | | | | | |
| Cr>5mg/dl | 154 | 55 | 7.913 (4.491-13.943) | 3.161 (0.216-8.220) | 0.18 |
| Cr1.3–4.9 mg/dl | 45 | 69 | 1.843 (1.006-3.378) | 1.017 (0.378-2.733) | 0.974 |
| Cr 0.71.2 mg/dl | 23 | 65 | 1.00 | 1.00 | |
| Inotropic used | | | | | |
| Yes | 164 | 74 | 4.394 (2.892-6.676) | 3.532 (0.753-7.115) | 0.2 |
| No | 58 | 115 | 1.00 | 1.00 | |
| Sedation used | | | | | |
| Yes | 98 | 132 | 0.341 (0.227-0.513) | 0.368 (0.183-0.740) | 0.005** |
| No | 124 | 57 | 1.00 | 1.00 | |

NB**; =P<0.05, 1.00 =reference AOR=Adjusted odd ratio; CI; Confidence Interval.

**Multi-organ dysfunction syndrome (MODS)**

This finding showed MODS is found to be increased the likelihood of mortality more than threefold **(AOR = 3.48; 95% CI: 1.47–7.82).**

**Sedation**

According to this study, sedation use was another significant factor that determines the outcome of patients with MV. It is revealed that sedation was associated with **lower odds of mortality with (AOR = 0.36; 95% CI: 0.18–0.74).** This protective relationship may reflect improved comfort and adaptation with the ventilator.

## Discussion

The number of critically ill patients requiring mechanical ventilation (MV) is rising globally, including in Ethiopia, due to increases in trauma, infectious diseases, and non-communicable diseases (NCDs) [3]. This study, conducted at JMC's AICU from 2019 to 2023, provides significant insights into the socio-demographics, clinical characteristics, management practices, and outcomes of patients who received mechanical ventilation. The findings underscore the critical factors associated with mortality and recovery, offering valuable implications for clinical practice and future research.

In the current study, the overall mortality rate among mechanically ventilated patients was 54% with a 95% CI (49.4, 58.6), This finding is comparable to the findings of other studies conducted in resource-limited settings, such as Egypt (54.7%) [26] and Millennium Medical College in Ethiopia (57.1%) [27]. These similarities are likely due to the comparable demographics and socioeconomic conditions of the patient populations, as well as the similarity in admission diagnoses, particularly respiratory failure, which was the most common indication for MV in both studies.

However, the mortality rate in this study was lower than that reported in Uganda (73.5%) [28] and Kenya (60.7%) [29]. A potential explanation for this discrepancy might be that the facilities at Jimma Medical Center are relatively better equipped, and the critical care professionals in the ICU may be more skilled than those in the rural hospitals in these other countries.

Conversely, the mortality rate in our study was significantly higher than the rates reported in middle- and high-income countries like Canada (18%) [18], Brazil (34%) [30], and Korea (36%) [31]. Delays in ICU admission, scarcity of resources, patients unable to pay the cost, since 84.7% died of ventilated patients who had no access to insurance in this study, and a lack of standardized ICU protocols in Ethiopia are likely causes of this disparity [14]. The mortality rate in this study was also higher than in studies conducted in southern India (42.1%) [32]. The highest mortalities in this study might be due to late referral to the ICU and admission problems (admitting unsalvageable patients due to lack of standardized protocol for ICU admission in this setting). Additionally, the rate of mortality in this study was higher than in Ayder Hospital in Ethiopia (28.6%) [3]. A possible justification for this discrepancy might be due to the high re-intubation rate (33.6%), and most of the re-intubated patients developed complications (61.3%) in this hospital, which was a risk for death in this study. However, in Ayder Hospital, only 10% of patients had re-intubations, of which only 35.3% had developed complications.

In this study, the mortality rate was higher in patients with respiratory disease (45%), which is similar to a study conducted in western India [33]. The majority of deaths of patients were those who were admitted from emergencies, which was a similar study conducted in western India [33] and Brazil [27]. However, a study conducted in Sub-Saharan Africa reveals that the majority of deaths of patients were those from medical wards (56.2%) [14]. This difference might be because the majority of ICU admissions were from emergencies, and different patients referred to this hospital as emergencies.

Among dead patients, the majority of age was found between 18 and 40 years, which was similar to the studies done in southern India [32], Nigeria [34], and Sub-Saharan Africa [14], but studies conducted in western India [33] and Korea [34] found older patients to be over 60 years old. This might be different due to the most common ICU admission in Ethiopia related to an emergency car accident vulnerable to the young population.

Acute respiratory distress syndrome (ARDS) was the main cause of death for patients who were on mechanical ventilation in this study (43.7%), which is similar to a study conducted in Western India [33]. Other studies done in Southern India [32], Nigeria [34], and sub-Saharan Africa [14] also show that acute respiratory distress syndrome was the main cause of death for those patients who were on mechanical ventilators, which ranged from 38.6% to 86.07%. Nevertheless, the study done in Egypt revealed cardiac disease as the predominant cause of death. This discrepancy might be due to the type of ICU they had—the cardiac ICU, the respiratory ICU—and the different categories of disease settings admitting patients based on specific diseases in each particular ICU.

The study identified several factors significantly associated with increased mortality, including prolonged hospital stay before intubation, re-intubation, the presence of comorbidities, the development of complications, MODS, and sedation.

Patients who stayed in the hospital for more than five days before intubation had three times higher odds of mortality. The mean duration of hospital stays before intubation (5.3 days) and the length of stay on mechanical ventilation (10.1 days) indicate the prolonged and complex care required for these patients. The use of invasive mechanical ventilation (IMV) in 96.4% of cases and endotracheal tube (ETT) as the primary airway access method (91.5%) is different from standard critical care practices [4,35]. This finding suggests that delays in initiating mechanical ventilation may contribute to worse outcomes; possibly discrepancy might be due to the progression of underlying disease and the development of complications [14].

Re-intubation is often a marker of failed initial weaning attempts [31]. It may indicate underlying issues such as unresolved primary illness or new complications like endotracheal tube blockages, accidental extubating, and self-extubating, all of which were common in this study. Patients who underwent re-intubation had a nine-fold increase in the odds of mortality compared to those who were not re-intubated. This result is consistent with findings from studies conducted in Korea [36], India [32], Brazil [30], China [18], and Sub-Saharan Africa [14]. Re-intubation increases the risk of complications, particularly VAP, as it is an invasive procedure that increases the likelihood of infection. The high re-intubation rate at JMC highlights the need for better monitoring of ventilated patients and more robust protocols for preventing unplanned extubation.

The presence of comorbidities and the development of complications were both strongly associated with increased mortality. A significant proportion of ventilated patients in this study had comorbidities, with congestive heart failure (28.8%), hypertension (20%), chronic kidney disease (12%), and diabetes mellitus (10%) being the most common. These comorbidities were directly associated with higher mortality rates. This finding aligns with studies conducted in India [33] and China [18].

In Ethiopia, the high burden of chronic diseases is often undiagnosed or poorly managed in the general population, especially in rural areas [14]. This increases the vulnerability of patients requiring mechanical ventilation, as they are more prone to complications like multi-organ dysfunction syndrome (MODS), which was seen in 76.9% of the patients in this study. These findings highlight the importance of early identification and management of comorbid conditions and vigilant monitoring for complications during mechanical ventilation.

In this study, 61.3% of patients developed complications during ventilation, with ventilator-associated pneumonia (VAP), hospital-acquired pneumonia (HAP), electrolyte imbalances, and bedsores being the most common. These complications significantly contributed to the high mortality rate, consistent with studies from Egypt [5] and Sub-Saharan Africa [14]. However, this result was higher than in Brazil [30,37], and China [38]. The high rate of complications can be attributed to suboptimal infection control practices and a shortage of critical care nurses in the ICU at JMC. This is a common issue in LMICS where healthcare workers may lack the necessary training in ICU protocols and infection prevention [39].

The development of MODS was associated with a three-fold increase in mortality risk. Multi-organ dysfunction syndrome (MODS) was observed in 76.9% of patients, with renal failure (36.7%) and cardiac failure (25.3%) being the most common. This is consistent with findings in other low-middle-income countries [9,14,26]. However, the rates were higher than in developed nations such as Brazil [30] and India [38], highlighting the severity of illness in patients admitted to the ICU at JMC. The

development of MODS is often a consequence of delayed ICU admission and inadequate management of complications, which is common in resource-limited settings like Ethiopia. The possible justification for this might be the lack of specialized healthcare workers such as respiratory therapists, and limited diagnostic facilities often result in delayed treatment for complications like MODS. Investments in specialized training and diagnostic capabilities are essential for improving outcomes.

According to this study, sedation use was shown to be an important factor significantly associated with mortality in mechanically ventilated patients. Sedation use was found to decrease the odds of mortality by 63.2%, as it prevented agitation and self-extubating, which could lead to traumatic injuries and respiratory failure [ 22,40]. However, excessive or prolonged sedation has been linked to higher mortality in other studies [41]. The judicious use of sedation, guided by protocols such as the Richmond Agitation-Sedation Scale (RASS), is crucial in LMICs to balance the risks and benefits of sedation in critically ill patients [42]. Continuous monitoring and appropriate titration of sedation could reduce the complications associated with mechanical ventilation [31]. This finding is supported by previous studies in Sub-Saharan Africa [14]. However, studies from other regions, such as Egypt, have shown that prolonged sedation can increase the risk of mortality due to delayed extubating [5]. The possible justification for this controversial practice is likely at JMC, where the controlled use of sedation is guided by the Richmond Agitation-Sedation Scale, which helps healthcare professionals maintain an optimal sedation level that prevents agitation without delaying extubating.

The strengths of this study provide a comprehensive analysis of the clinical outcomes of mechanically ventilated patients in the AICU over five years, making it the first of its kind in the southwest region of Ethiopia. By examining a diverse cohort of patients across different units, the research enhances the generalizability of the findings.

Despite, this, study has some limitations that should be acknowledged. Firstly, the retrospective design means that direct intervention or real-time observation of patients was not possible, which may have limited the collection of some important variables relevant to the study. Additionally, those who had requested against medical advice could not be tracked with poor prognoses until they passed away or recovered, making it unable to assess all outcomes of ventilated patients.

## Conclusion and recommendations

### Conclusion

This study demonstrated that mortality among mechanically ventilated adult patients at Jimma Medical Center remains alarmingly high, reflecting the complex clinical conditions of patients as well as systemic challenges common in resource-constrained ICUs. Key determinants of mortality were delayed intubation, re-intubation, comorbidities, complications during ventilation, and the presence of multiple organ dysfunction syndromes (MODS). These findings pinpointed gaps in critical care delivery and reinforce the need for timely recognition of respiratory failure, structured decision-making for intubation, and meticulous monitoring of mechanically ventilated patients. Although resource limitations remain a significant barrier, targeted interventions such as improving pre-ICU triage, early initiation of care, applying standardized ventilator care packages, and strengthening airway management practices may substantially reduce avoidable deaths. The protective association observed with sedation use suggests that supportive measures promoting ventilator synchrony and physiologic stability should be incorporated thoughtfully into patient management.

### Recommendations

Based on the findings of this study, the following recommendations can help improve clinical outcomes for mechanically ventilated patients at Jimma Medical Center (JMC) and similar resource-limited settings.

**For hospital administrators:** It is better to arrange different trainings for the AICU health professionals regarding complications related to mechanical ventilators.

**For hospital health professionals**: It is urged that this hospital focus on patients who stayed more than 5 days before intubation after the hospital stay, timely screening patients, and early referral to the ICU, so that severity scores

can be computed and used in the ICU. Health professionals shall be skilled in full training to reduce the frequency of re-intubation and complications made before, during, and after intubation among the patients intubated for treatment. Also, appropriate use of sedation scales for agitated patients, such as the Richmond Agitation Sedation Scale (RASS), is advised.

**For Jimma Medical Center AICU**: Emphasis shall be given by using severity indicator scores to show special concern for patients with organ failure and previous comorbidities, to strengthen their complications related to the mechanical ventilation management protocol, to decrease the likelihood of patient intubation in the ICU, and to improve the quality of care since mortality is high. Also, this high mortality rate suggests an urgent need for extensive improvement in the protocol for ICU setup and special attention to re-intubation, comorbidity, complications, MODS, and delayed intubation to reduce the mortalities of ventilated patients.

**For researchers**: It is better to conduct a prospective study to find out the clinical outcome and its associated factors among mechanically ventilated patients. It will also be best if they add extra hospitals to their study to give better generalizations.

## Acknowledgments

First of all, we would like to express our sincere appreciation to Jimma University for giving us this chance to participate in the research activity. We would like to acknowledge Jimma University Medical Center, Shanan Gibe General Hospital, and their respective staff for their support while undertaking this study. Also, we are grateful to acknowledge our study subject who voluntarily participated in this study for providing the necessary information, and the data collectors for collecting the data carefully.

## Author contributions

**Conceptualization:** Abera Mulatu Uma.

**Data curation:** Abera Mulatu Uma, Segni Gemechu Hurisa, Mulu Kitaba Negawo, Sheka Shemsi Seid.

**Formal analysis:** Abera Mulatu Uma, Sheka Shemsi Seid.

**Methodology:** Abera Mulatu Uma, Shemsadin Ame, Segni Gemechu Hurisa, Mulu Kitaba Negawo, Sheka Shemsi Seid.

**Project administration:** Abera Mulatu Uma.

**Software:** Abera Mulatu Uma, Segni Gemechu Hurisa, Mulu Kitaba Negawo, Sheka Shemsi Seid.

**Supervision:** Shemsadin Ame, Segni Gemechu Hurisa, Sheka Shemsi Seid.

**Validation:** Abera Mulatu Uma, Shemsadin Ame, Sheka Shemsi Seid.

**Visualization:** Shemsadin Ame, Mulu Kitaba Negawo, Sheka Shemsi Seid.

**Writing – original draft:** Abera Mulatu Uma.

**Writing – review & editing:** Abera Mulatu Uma, Shemsadin Ame, Segni Gemechu Hurisa, Mulu Kitaba Negawo, Sheka Shemsi Seid.

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
