## [Decision Letter · Decision Letter 0]

28 Nov 2025

Dear Dr. Uma,

Thank you for submitting your manuscript to PLOS ONE. After careful consideration, we feel that it has merit but does not fully meet PLOS ONE’s publication criteria as it currently stands. Therefore, we invite you to submit a revised version of the manuscript that addresses the points raised during the review process.

We look forward to receiving your revised manuscript.

Kind regards,

Chiara Lazzeri

Academic Editor

PLOS ONE

Journal Requirements:

There is competing interest

5. Please provide a complete Data Availability Statement in the submission form, ensuring you include all necessary access information or a reason for why you are unable to make your data freely accessible. If your research concerns only data provided within your submission, please write "All data are in the manuscript and/or supporting information files" as your Data Availability Statement.

6. Please ensure that you refer to Figures 1 and 2 in your text as, if accepted, production will need this reference to link the reader to the figures.

7. Please upload a copy of Figure 6, to which you refer in your text on page 24. If the figure is no longer to be included as part of the submission please remove all reference to it within the text.

8. We note you have included a table to which you do not refer in the text of your manuscript. Please ensure that you refer to Tables 2 and 3 in your text; if accepted, production will need this reference to link the reader to the Tables.

Reviewers' comments:

Reviewer's Responses to Questions

**Comments to the Author**

1. Is the manuscript technically sound, and do the data support the conclusions?

Reviewer #1: Yes

Reviewer #2: Partly

2. Has the statistical analysis been performed appropriately and rigorously?

Reviewer #1: Yes

Reviewer #2: Yes

3. Have the authors made all data underlying the findings in their manuscript fully available?

Reviewer #1: Yes

Reviewer #2: Yes

4. Is the manuscript presented in an intelligible fashion and written in standard English?

Reviewer #1: No

Reviewer #2: No

Reviewer #1: Comments to authors

1. Abstract section

Methods part.

It is not a ‘retrospective case control study’. It is ‘retrospective study’ from chart review (secondary data). Please cancel ‘case-control…’.

Source of adapted ‘data extraction tool’ not mentioned. Was it copied from a single source or adapted from other similar studies?

2. Introduction section

Line 68-75 and 75-79 is redundant, difficult to understand. Would you please re-write it again?

Line 137-151 is a ‘recommendation’ that should be mentioned under discussion part. Please cancel it under ‘Introduction section’?

3. Methods and materials

Study design and period.

The study type is ‘retrospective study’ chart review study. Please cancel …case-control study….

Why cross sectional study design was selected to determine sample size for the second objective (clinical determinants of outcome)?

On data analysis part, What was the percentage of missing data?, what type of diagnostic imputation method was used to manage missing values?, What was the VIF value to evaluate testing for multicollinearity?

Under ethical consideration, why did you expose data for the medical director who was not the investigator? Breach in confidentiality?

Under dissemination of results, Use past tense, as it was submitted for publication and possibly submitted for responsible bodies in the country.

4. Results

There is improper labeling of tables and figures. Please revise it.

Where is table-1?

Line 395: See fig3 but it was fig 2

Define under foot note for ‘others’ for medical diagnosis under table: Admission diagnoses

Line-418: fig 3 is less contributory and better be omitted, and can be only mentioned in texts. The fig doesn’t add any new information.

Write foot note for ‘others’ for inotrope drug used for table: Management-related factors

Line-488 and line-493 mislabeled tables.

Line 486-487: Out of the total number of mechanically ventilated patients, two hundred twenty-two (54%) with 95% CI (49, 24, 58.6) died,

… 54% (CI: 49,24,58.6)? not clear. Is it 49.2 to 58.6%?

5. Discussions section

Line-539: Mention the confidence interval for mortality rate of 54% (CI: ??-??), which is important for comparison with other studies?

Line 546-548: Please explain in other words. I doubt of the justification given.

Justification of the determinants of outcome for mechanically ventilated patients requires better reasoning. Please read in detail regarding reasoning for outcome determinants.

Recommendation should focus based on the study, and be directed to health sectors in resource limited settings.

6. References

Please revise the references if correctly labeled based on journal style.

Reviewer #2: Reviewer comments

I'm grateful for the opportunity to review your manuscript. The study is conducted on the relevant topic, and the findings of the study are very important to the initiatives of quality patient care enabling the healthcare professionals to make informed decision based on the evidence from the studies like this. However, the manuscript lacks proper articulation and synthesis making it scientifically less sound. Therefore, IT Needs major revision.

Abstract: In the background section the authors listed possible outcomes of MV, which is more than two outcomes, so why you use bivariate analysis while having more than two outcomes? Either describe how you classify your outcomes in the methods section.

in the result section, be consistent with the decimal points and parameters. eg. a

multi-organ failure (AOR = 3.477, 95% CI: 1.471-7.8.215, p< 0.005) in this case the authors used three decimal places and P-value inconsistent with the other.

In the conclusion and recommendation section, the recommendations suggested by the authors is generic; be specific with your recommendations based on your findings.

Keywords: should be italicized

Introduction

The authors addressed the scientific background, magnitude and severity of the problem under study. Additionally, the authors also conducted literature reviews and tried to identify the research gap in the area. However, the following concerns need to be addressed:

1. Generally, the introduction lacks organization and synthesis of ideas, making the section too long. Summarization, organization and synthesis of the ideas in the section would benefit the manuscript.

2. The authors elaborated the relevant literature, which is positive, however it’d better if it can be summarized and organized into two core paragraphs.

3. Editorial: 1. Long unclear sentences (Line 81-85), incomplete sentences (Line 87-90), wrong wording (eg. Factories),

4. Abbreviations are used without being explained on their first use. Eg ICU, MV, AICU…

5. The authors identified limited studies in Ethiopia and important variables missing in the prior studies, making it difficult to generate compelling evidence. However, the authors didn’t clearly describe what variables the current study addressed. In addition, the authors mentioned that there is no study conducted in the current setting, why do you think we can’t generalize the other studies for your setting?

6. The last four paragraphs can be summarized into one paragraph. (Line 124-151)

7. In conclusion major editorial revision required with grammar, sentence construction and wording

Methods and Materials. Use journal guidelines while naming titles and subtitles.

1. Study area: add study period to the subtitle, remove study period from study design. The study was conducted between April – May 2024. Was this the data collection period or study period? For retrospective study what is the study period?

2. Study population: why do the authors select the patients admitted from 2019 to 2023?

3. Your exclusion criteria need clarification. Why do you exclude these patients? Referral could also be considered as one of the outcomes of the patient. Is it possible to exclude because of illegible handwriting? What about incomplete patient charts?

4. Sample size estimation and sampling procedure: You don’t have to copy everything in your main document into your manuscript. Summarize into one paragraph.

5. Variables: your variables look overlapping eg. Length of hospital stay VS Length of stay in the hospital before intubation. You need to write these abbreviations in expanded form VCV, PCV, SIMV, MODS and CPAP. How do you think residence affects the outcome of mechanical ventilation?

6. Operational definition: Operational definition is what makes your variables measurable. Therefore, limit them to only variables in your study. There are many unnecessary statements in your operational definition. Operational definition is not describing about the subtheme of your variables. so, remove them. Normal and abnormal ranges of different indicators, just limit to only important variables.

7. Generally, this manuscript looks immature and is the direct copy of master's thesis, so I recommend the authors to revise the manuscript according to the journal guideline.

Results

1. Line 391-392, You are not expected to report about the patients not included in your study.

2. Figure 2: You may have used another visualization which would better explain the data. eg. Stacked bar graph.

3. Line 407, what does the number 5.2.1 indicate? The subtitles can be rewritten as clinical characteristics of the patient. Regarding table 2, I couldn’t in which type of table another name of column could be inserted midway the table.

4. The authors presented their results both in words and diagrams. However, it lacks synthesis and organization. How do the authors define comorbidity? What is the demarcation between reason of admission, comorbidity, MODs? All the variables presented in the result sections should be clearly operationalized in the methods section.

5. The result indicating study outcome is presented, but the authors used an informal diagram (broken pie chart). Editorial 95% CI (49, 24, 58.6)

6. The authors classify the length of stay as <5 days and > 5 days, what is your reference for the classification?

7. Please limit your variables to only the relevant ones

Discussion

1. The first paragraph is good, however in your comparison the authors compared studies conducted in Egypt and Ethiopia at the same time, the rationale the authors provided might not work for both.

2. Line 545-548, do you have evidence that JMC is better equipped and has well-skilled professionals than Uganda and Kenya? Please provide compelling evidence

3. Tailor your discussion to your objectives, be consistent with your justification

Remove “indications for clinical practices” section incorporate the idea into the discussion.

Remove strengths and limitations section, Conclusion and recommendation could be summarized into the last paragraph of discussion.

.

Reviewer #1: **Yes:**Abilo Tadesse, MDAbilo Tadesse, MDAbilo Tadesse, MDAbilo Tadesse, MD

Reviewer #2: No

You may also use PLOS’s free figure tool, NAAS, to help you prepare publication quality figures: https://journals.plos.org/plosone/s/figures#loc-tools-for-figure-preparation

---

## [Author Response · Author response to Decision Letter 1]

12 Jan 2026

to the best of our understanding we try to address for all comments given by editors and reviewers. besides, our suggestion we look forward to your constructive feedbacks.

---

## [Editor Report · Decision Letter 1]

13 Jan 2026

Clinical Outcomes and Associated Factors Among Mechanically Ventilated Patients in Adult Intensive Care Unit at Jimma Medical Center, Southwest Ethiopia, 2024.

PONE-D-25-52527R1

Dear Dr. Uma,

We’re pleased to inform you that your manuscript has been judged scientifically suitable for publication and will be formally accepted for publication once it meets all outstanding technical requirements.

Kind regards,

Chiara Lazzeri

Academic Editor

PLOS One
---

## [Editor Report · Acceptance letter]

PONE-D-25-52527R1

PLOS One

Dear Dr. Uma,

I'm pleased to inform you that your manuscript has been deemed suitable for publication in PLOS One. Congratulations! Your manuscript is now being handed over to our production team.

Kind regards,

on behalf of

Dr. Chiara Lazzeri

Academic Editor

PLOS One